# Whole-Genome Sequencing-Based Profiling of Antimicrobial Resistance Genes and Core-Genome Multilocus Sequence Typing of *Campylobacter jejuni* from Different Sources in Lithuania

**DOI:** 10.3390/ijms242116017

**Published:** 2023-11-06

**Authors:** Jurgita Aksomaitiene, Aleksandr Novoslavskij, Mindaugas Malakauskas

**Affiliations:** Department of Food Safety and Quality, Faculty of Veterinary Medicine, Lithuanian University of Health Sciences, Tilzes Str. 18, LT-47181 Kaunas, Lithuania; aleksandr.novoslavskij@lsmu.lt (A.N.); mindaugas.malakauskas@lsmu.lt (M.M.)

**Keywords:** antimicrobial resistance, multidrug resistance, whole-genome sequencing, *Campylobacter jejuni*, cgMLST, antimicrobial resistance genes

## Abstract

*Campylobacter jejuni* is known as one of the main causative agents of gastroenteritis in humans worldwide, and the rise of antimicrobial resistance (AMR) in *Campylobacter* is a growing public health challenge of special concern. Whole-genome sequencing (WGS) was used to characterize genetic determinants of AMR in 53 *C. jejuni* isolates from dairy cattle, broiler products, wild birds, and humans in Lithuania. The WGS-based study revealed 26 *C. jejuni* AMR markers that conferred resistance to various antimicrobials. Genetic markers associated with resistance to beta-lactamases, tetracycline, and aminoglycosides were found in 79.3%, 28.3%, and 9.4% of *C. jejuni* isolates, respectively. Additionally, genetic markers associated with multidrug resistance (MDR) were found in 90.6% of *C. jejuni* isolates. The WGS data analysis revealed that a common mutation in the quinolone resistance-determining region (QRDR) was R285K (854G > A) at 86.8%, followed by A312T (934G > A) at 83% and T86I (257C > T) at 71.7%. The phenotypic resistance analysis performed with the agar dilution method revealed that ciprofloxacin (CIP) (90.6%), ceftriaxone (CRO) (67.9%), and tetracycline (TET) (45.3%) were the predominant AMR patterns. MDR was detected in 41.5% (22/53) of the isolates tested. Fifty-seven virulence genes were identified in all *C. jejuni* isolates; most of these genes were associated with motility (n = 28) and chemotaxis (n = 10). Additionally, all *C. jejuni* isolates harbored virulence genes related to adhesion, invasion, LOS, LPS, CPS, transportation, and CDT. In total, 16 sequence types (STs) and 11 clonal complexes (CC) were identified based on core-genome MLST (cgMLST) analysis. The data analysis revealed distinct diversity depending on phenotypic and genotypic antimicrobial resistance of *C. jejuni*.

## 1. Introduction

*Campylobacter jejuni* is a leading bacterial pathogen that causes acute gastroenteritis in humans and is found in the gastrointestinal tracts of most animals [1,2]. Most often, infection is acquired through the consumption of raw or undercooked poultry, unpasteurized milk, or contaminated water. Other potential sources of human campylobacteriosis include the environment, wild birds, and contact with animals [3,4]. While most *Campylobacter* infections are self-limiting and do not require antibiotic treatment, severe and prolonged cases should be treated with antimicrobials, especially in children, the elderly, and people with compromised immune systems [5,6]. Antimicrobial agents considered as the primary options for the treatment of *Campylobacter* infections are macrolides (erythromycin) and fluoroquinolones (ciprofloxacin). However, in some cases, tetracyclines and gentamicin, used to treat systemic infection, might be also considered as an alternative therapeutic option [7,8]. Antimicrobial resistance is a major public health issue which requires scientists to focus on AMR mechanism detection [9,10]. Whole-genome sequencing (WGS)-based detection of bacterial resistance is one of the most specific methods for identifying genes responsible for AMR [11]. WGS-based methods have been improved in recent years and currently can identify AMR genes and predict the AMR of *C. jejuni* with a high degree of accuracy (up to 99%) [8,12]. WGS data enable scientists to identify and analyze the AMR mechanisms, the different mobile genetic elements (MGEs), bacterial genes, and their mutations that play a critical role in resistance development, particularly in MDR pathogens [8,13]. Along with the potential to predict AMR more efficiently, WGS also gives the opportunity to perform the whole-genome multilocus sequence typing (wgMSLT) following different schemes [14].

The increase in the rate of antimicrobial resistance of *Campylobacter* strains originating from food of animal origin is noted during the last years [15,16,17]. Antimicrobial resistance in *C. jejuni* is frequently linked to single-point mutations in the bacterial genome, while the main mechanism for acquiring antibiotic resistance is believed to be horizontal gene transfer between bacterial isolates [18,19]. The gene mutations reduce or eliminate antibiotics’ ability to affect target sites, increasing antimicrobial resistance [20,21]. It was described that point mutation of the *gyrA* gene in the quinolone resistance-determining region (QRDR) is responsible for fluoroquinolone resistance in *C. jejuni* isolated from humans and animals [22,23]. This mutation is associated with a threonine change to isoleucine at position 86 (Thr86Ile), which encodes a subunit of the target enzyme [22,24]. *Campylobacter* resistance to β-lactams is mediated by a variety of mechanisms, including enzymatic inactivation by chromosomally encoded β-lactamases and reduced antibiotic uptake due to changes in outer membrane porins [6,25]. The point mutations regulating the expression of *blaOXA-61* and *blaOXA-184* genes are also linked to *j* high-level resistance to β-lactams [26,27].

CmeABC and CmeDEF multidrug efflux systems are responsible for *C. jejuni* resistance to multiple antibiotics including fluoroquinolones, erythromycin, tetracycline, chloramphenicol, and ampicillin [28,29]. Additionally, CmeABC multidrug efflux pumps can operate on antibiotic targets synergistically with spontaneous mutations of different resistance genes conferring high-level resistance to different antimicrobials [25,30]. Many bacterial species, including *Campylobacter*, harbor the aminoglycoside resistance genes, and enzymatic alteration of aminoglycosides by aminoglycoside-modifying enzymes is the main mechanism of resistance to these antibiotics. Recently, a high prevalence and predominance of the aminoglycoside resistance genes including *aac(6′)-Ie/aph(2″)-Ia*, *aph(2″)-If*, and *aph(2″)-Ig* were reported in *Campylobacter* [31,32,33].

The DNA-based methods (like PCR or mPCR assays) are considered as the most viable methods in molecular diagnostics used to detect genes and point mutations related to AMR. However, these approaches provide very limited information on AMR gene presence and mutations within the context of the genome of bacteria to gauge their likelihood of transfer across genomes. The advantages of the study of AMR, based on WGS analysis, include the ability to predict or provide the full complement of resistance genes present for several isolates as well as the characterization of mutations that might confer antimicrobial resistance. Moreover, such investigation gives the ability to perform whole-genome sequencing-based typing, including core-genome multilocus sequence typing (cgMLST), of isolates.

This study aimed to evaluate the antimicrobial resistance phenotypes and genetic diversity associated with WGS-based profiling of antimicrobial resistance genes of *C. jejuni* isolated from broiler products, cattle, wild birds, and human feces.

## 2. Results and Discussion

### 2.1. Antimicrobial Resistance Phenotypes

In total, 53 *C. jejuni* isolates were screened for antimicrobial resistance, and 90.6% of isolates were resistant to at least one out of five tested antibiotics.

Most often, *C. jejuni* were resistant to ciprofloxacin (90.6%), ceftriaxone (67.9%), and tetracycline (45.3%). Only some *C. jejuni* isolates were resistant to gentamicin (3.8%) and erythromycin (1.9%) in our study. Similar findings on the high level of resistance to ciprofloxacin and tetracycline in *C. jejuni* of different origins were also reported in previous studies [34,35]. We found that resistance to tetracycline and ceftriaxone in *C. jejuni* isolates from dairy cattle was 100%, in comparison to 20% and 45% resistance in bacteria isolated from human feces, respectively. The level of AMR in *C. jejuni* can vary depending on the source of isolation, such as from food, animals, or humans. The variability depending on the isolation source also has been demonstrated in several previous studies conducted in Norway, Italy, China, and Spain [27,36,37,38].

In total, 41.5% of the *C. jejuni* tested were confirmed as multidrug-resistant (MDR), especially isolated from dairy cattle (100%) and broiler products (50%). The MDR rate reported in this study showed similar results to those obtained in studies conducted in Spain [39]. However, significantly higher MDR resistance was found in studies conducted in Kenya, where MDR for *C. jejuni* was reported at 94.6% [40].

The TET + CIP + CRO antimicrobial resistance profile was predominant and confirmed in 37.7% (Figure 1a). Importantly, two (3.8%) *C. jejuni* isolates originating from wild birds were confirmed as extensively drug-resistant (XDR). Obtained data indicate that XDR *C. jejuni* strains circulate in wildlife and can pose a significant threat to public health, as it can be challenging and sometimes impossible to treat campylobacteriosis infections with antibiotics [41]. The possible sources could be the environmental transmission of antibiotic-resistant bacteria and resistance genes, as wild birds frequently come into contact with various environmental reservoirs, including water bodies, soil, and other wildlife [42,43,44].

Evaluation of the resistance profiles distance matrix showed that *C. jejuni* isolates from human feces exhibited a high degree of similarity with isolates from broiler products (Figure 1b). These findings suggest a link between antimicrobial resistance in human populations and the use of antibiotics in animal production [45]. The similarity/dissimilarity of resistance patterns between *C. jejuni* isolates was assessed by calculating the pairwise matrix distance shown in Figure 1.

### 2.2. Genomic Characterization

The assembled genome size per strain ranged from 1.6 to 1.86 Mb, and the average G + C content of all genomes was 30.32%. The average N_50_ of assemblies was 273 Kb. The average number of contigs was estimated to vary between 11 and 105 per isolate. The pan-genome analysis examined the gene diversity content and defined a total of 4204 genes, of which 1215 were assigned as core genes and 2989 were assigned as accessory genes, representing 28.9% and 71.1%, respectively. The accessory genome consisted of 56 softcore genes (95% ≤ isolates ≤ 99%), shell genes (15% ≤ isolates ≤ 95%), and cloud genes (0% ≤ isolates ≤ 15%).

A minimum spanning tree (MST) was defined for 53 *C. jejuni* genomes using 1369 targets (1,328,103 bases) for cgMLST and 220 accessory targets (145,326 bases). The seed genome for selecting target genes belonged to *Campylobacter jejuni* NCTC11168 (NZ_LS483362.1) strain and required a minimum length filter ≥50 bases, homologous gene filter with overlap ≥100 bp, identity ≥90.0%, and gene overlap filter with >4 bases. The genomes there were grouped based on sequence type using a cluster threshold of ≤6 allelic differences. Out of the 53 *C. jejuni* genomes, three clusters were detected. The largest cluster (cluster 1 in Figure 2) was composed of three isolates of the same ST-5 (CC353), however of different origins (two isolates from human feces and one isolate from broiler products). Meanwhile, cluster 2 was composed of two *C. jejuni* isolates assigned to CC21 ST-21 of the same origin (cattle). Moreover, these isolates were confirmed as multidrug-resistant with a TET + CIP + CRO resistance profile (cluster 2 in Figure 2). Cluster 3 was composed of two *C. jejuni* isolates assigned to CC353 ST-5 of the same origin (poultry) (cluster 3 in Figure 2). The SeqSphere analysis of *C. jejuni* revealed 16 distinct sequence types classified into 11 distinct clonal complexes. The MLST profiles were compared using Euclidean distance, which resulted in calculating a matrix of genetic distances between all isolate pairs. This measure of distance value indicated the extent of difference between the two profiles, and higher values indicated greater genetic distances. Furthermore, the genetic relatedness was calculated using the Jaccard coefficient based on the presence or absence of alleles at each locus. The *C. jejuni* isolates of the sequence type (ST-5) were assigned to the same cluster, which suggests that they are highly related and may have a common ancestor. The fact that *C. jejuni* isolates of the sequence type (ST-5) are in the same cluster means that they share similar genetic profiles and likely have similar phenotypic characteristics. Hierarchical clustering reflecting the genetic relatedness of *C. jejuni* isolates is shown in Figure 3.

### 2.3. Analysis of Genotypes and Phenotypes of C. jejuni Isolates

Further analysis revealed that ST-5 predominated among 53 *C. jejuni* isolates (24.5%), followed by ST-21 (13.2%), ST-50 (11.3%), and ST-6411 (11.3%). *C. jejuni* isolates assigned to ST-5 were dominant among *C. jejuni* isolated from broiler products and were resistant to ciprofloxacin and ceftriaxone. Additionally, ST-21 (11.3%), ST-4447 (7.5%), and ST-6411 (7.5%) were the predominant sequence types among multidrug-resistant (MDR) *C. jejuni* strains. These findings support the opinion that specific sequence types, such as ST-6411, are associated with MDR resistance [46,47]. Furthermore, the significant predominance of ST-6411 in both isolate distribution and MDR patterns was also reported by Wieczorek et al. [48]. WGS analysis of 53 *C. jejuni* strains identified 26 AMR genes and resistance-associated point mutations which conferred resistance to a variety of antimicrobials, including tetracycline, aminoglycosides, streptothricin, β-lactams, and the drug efflux pump family. Resistome analysis revealed that genes *cmeA*, *cmeB*, *cmeC*, *cmeD*, *cmeE*, and *cmeR* (median = 98.4%) belonging to multidrug efflux system were the most prevalent in all *C. jejuni* genomes.

In total, 79.3% (n = 42) of the *C. jejuni* genomes harbored at least one of the identified β-lactams genes. The *blaOXA-65* gene was identified in 23 *C. jejuni* isolates (43.4%), with 21 isolates (41.5%) coharboring *blaOXA-61*, *blaOXA-193*, and *blaOXA-450* genes. In total, four *C. jejuni* isolates harbored one *blaOXA-184*, *blaOXA-446*, and *blaOXA-447* genes (Figure 4). The β-lactam resistance in *Campylobacter* is mediated by various mechanisms, including enzymatic inactivation by chromosomally encoded β-lactamases or decreased uptake due to changes in outer membrane porins and efflux [6,25]. Most often, *Campylobacter* are considered to be resistant to β-lactam antimicrobial agents and narrow-spectrum cephalosporins due to their limited ability to bind to penicillin-binding proteins (PBPs) and their low permeability. Meanwhile, *Campylobacter* resistance to other β-lactam antibiotics, such as amoxicillin and ampicillin, is explained by the production of β-lactamase, which inactivates β-lactam antibiotics by hydrolyzing the structural lactam ring [31]. Based on WGS analysis, three genes associated with aminoglycoside resistance, *aph(2″)-Id*, *ant(6)-Ia*, and *aph(3′)-III*, were identified in 9.4% (n = 5) of *C. jejuni* genomes. It is known that enzymatic drug modification and mutations at ribosomal binding sites are two mechanisms of aminoglycoside resistance in *Campylobacter* [38]. The streptothricin acetyltransferase gene *sat4* was detected in five (9.4%) *C. jejuni* isolates in our study. This horizontally transferable *sat4* gene encoding resistance to streptothricin’s was also previously detected in *Campylobacter* isolates of different animal and clinical sources [49,50]. The present study’s results revealed that some ST-21 (CC21) isolates, associated with MDR and assigned to the aminoglycoside resistance gene cluster, coharbored streptothricin and β-lactams resistance genes but lacked the *cmeB* gene. (Figure 4).

WGS data analysis revealed that *C. jejuni* isolated from broiler products and assigned to ST-21 (CCNA) harbored the majority of AMR genes (15 out of 26 genes tested). Data analysis also revealed that tetracycline resistance coding gene *tetO* was found in 15 *C. jejuni* genomes. Of note, one *C. jejuni* isolate from human feces (CJ13) assigned to ST-446 (CC446) with the TET+CIP+CRO resistance profile harbored *tetO*, and *blaOXA-184* genes and multidrug efflux system genes like *cmeA*, *cmeB*, *cmeC*, *cmeD*, *cmeE*, and *cmeF* (Figure 4). Moreover, CJ13 isolate had G → A change in the 250 position of the *cmeR* gene. WGS data analysis revealed 13 amino acids changes in the *gyrA* gene. The R285K (854G > A), transition in the quinolone resistance-determining region (QRDR) was the most frequent and identified in 42 (86.8%) of the tested *C. jejuni* isolates. The most frequent *gyrA* point mutations included the A312T (934G > A) and T86I (257C > T) transitions, detected in 44 (83%) and 38 (71.7%) of *C. jejuni* isolates, respectively. The T86I mutation leads to increased resistance to FQs; moreover, the point mutations at multiple positions of the DNA gyrase A region can cause high-level resistance to FQs [51,52]. Additionally, other identified substitutes linked to fluoroquinolone resistance like T665S (197C > G) and S22G (64A > G) were found in in 58.2% and 51.2% of *C. jejuni* isolates, respectively. Of note, 27 *C. jejuni* isolates possessed a cytosine (C) → thymine (T) nonsense mutation in 2587 bp (Q863*) upstream of the stop codon (Figure 5).

In total, 13 point mutations in the *cmeR* inverted repeat (IR) region were found after testing 53 *C. jejuni* isolates. *cmeR* gene is a transcriptional regulator in the *tetR* family, and the substitution in *cmeR* causes overexpression of CmeABC [28]. We found that the most frequent substitutions in *cmeR* were 431G > A (88.3%) and 619A > G (55.8%); there, glycine in the 144 position was changed by aspartic acid, and serine in the 207 position was changed by glycine, respectively. The *cmeR* substitutions D52N (154G > A), G53S (157G > A), A98T (292G > A), I115V (343A > G), and L184V (550C > G) were identified only in isolates from wild birds belonging to CC952 (ST-6228, ST-2111). The combined D121N (361G > A) and E159K (475G > A) amino acid changes were detected in MDR isolates isolated from cattle assigned to ST-21 CC21.

The distribution and functional roles of virulence and survival genes among the 53 *C. jejuni* isolates are shown in Figure 6. A total of 117 genes associated with various functions, encompassing adherence, motility, invasion, stress response, capsule synthesis, oxidative stress resistance, cytolethal distending toxin production (CDT), lipooligosaccharide synthesis (LOS), lipopolysaccharide synthesis (LPS), regulatory and transport systems, and the type IV secretion system (T4SS) production were identified.

Fifty-seven virulence genes were identified in all *C. jejuni* isolates; most of these genes were associated with motility (n = 28) and chemotaxis (n = 10). Additionally, all *C. jejuni* isolates harbored virulence genes related to adhesion (*cadF*, *jlpA*, *pebA*), invasion (*ciaB*, *ciaC*, *eptC*), LOS (*gmhA*, *gmhB*, *hldD*, *hldE*), LPS (*htrB*, *waaC*, *waaT*), CPS (*kpsC*, *kpsD*, *kpsF*), transportation (*kpsS*, *kpsT*), and CDT (*cdtC*). All *C. jejuni* isolates belonging to ST-21 (CC21) were found to possess the lipooligosaccharide synthesis *wlaN* virulence gene, which is known as a gene responsible for the ganglioside mimicking Guillain-Barré syndrome [53].

Among all 53 isolates, only one wild bird isolate (CC952 ST-2111) harbored seven virulence genes that belonged to T4SS. Almost all isolates from cattle (CC21 ST-21) with MDR profile carried the same virulence profile, harboring 99.2% of virulence genes.

## 3. Materials and Methods

### 3.1. Study Isolates

In total, 53 *C. jejuni* isolates (from cattle (n = 6), wild birds (n = 13), broiler products (n = 14), and human feces (n = 20)) from a bacterial culture collection of the Department of Food Safety and Quality of the Lithuanian University of Health Sciences were tested in this study. Isolates were collected over a one-year period from humans with clinical signs consistent with campylobacteriosis and confirmed *C. jejuni* infections, as well as from dairy cattle, poultry products (drumsticks and wings), and wild birds, including crows and pigeons. The isolates were stored at −80 °C in brain heart infusion broth (BHI) (Oxoid, Basingstoke, UK) with 30% glycerol (Stanlab, Lublin, Poland). The *C. jejuni* recovery was performed by plating the stocks on blood agar base No. 2 (Oxoid, Basingstoke, UK) supplemented with 5% defibrinated horse blood (E&O Laboratories Limited, Bonnybridge, UK) and further incubation under microaerophilic conditions (5% oxygen, 10% carbon dioxide, and 85% nitrogen) at 37 °C for 48 h.

### 3.2. Antimicrobial Susceptibility Testing

All bacteria isolates were tested with antimicrobial susceptibility to erythromycin (ERY), tetracycline (TET), gentamicin (GEN), ciprofloxacin (CIP), and ceftriaxone (CRO) (all from Sigma-Aldrich, Saint Louis, MO, USA). The agar dilution method was used to determine the minimum inhibitory concentration (MIC) in accordance with Clinical and Laboratory Standards Institute (CLSI) guidelines [54]. All isolates were cultured on Mueller–Hinton agar (Liofilchem, Roseto degli Abruzzi, Italy) plates at dilutions ranging from 0.25 to 256 µg/mL. For each individual *C. jejuni* isolate, 5 µL of approximately 1 × 10^7^ CFU/mL (OD_600_ = 0.1) bacterial suspension dissolved in phosphate-buffered saline (PBS) (E&O Laboratories Limited, Bonnybridge, UK) was spotted onto Mueller–Hinton agar plates containing the corresponding antimicrobial agent concentration. The inoculated Petri dishes were incubated for 24 h under microaerophilic conditions at 37 °C. The MIC value was defined as the lowest concentration that produces complete inhibition of *C. jejuni* growth. For quality control, the reference strain of *C. jejuni* ATCC 33560 was included. Isolates with confirmed resistance to three or more classes of antimicrobials were considered as multidrug-resistant.

### 3.3. Whole-Genome Sequencing and Data Analysis

Genomic DNA was extracted using a PureLink Genomic DNA Kit (Invitrogen, Carlsbad, CA, USA) and eluted in 50 L of sterile Mili-Q water. The quantity, concentration, and integrity of DNA were estimated using a Nanodrop 2000 spectrophotometer, a Qubit 3.0 fluorometer (Thermo Fisher Scientific, Waltham, MA, USA) with a dsDNA HS assay kit (Life Technologies, Eugene, OR, USA) and a 1% agarose gel, respectively. Genomic library construction was prepared using the Nextera XT DNA sample preparation kit (Illumina, San Diego, CA, USA) according to the manufacturer’s instructions and sequenced on an Illumina MiSeq platform (Illumina, San Diego, CA, USA). The quality of raw reads (2 × 150 bp paired end) was assessed using the FastQC v.0.11.9 tool [50], and the Trimmomatic v.0.38 tool [51] was used for the adapter trimming. Reads were de novo assembled into contigs using the SPAdes v.3.15.1 assembler [52]. The quality of the assembled sequences was assessed using the QUAST v.5.2 tool [53], discarding contigs of less than 300 bp. The annotation was performed using the Prokka v.1.14.6 software tool [55]. Ridom™ SeqSphere+ v.8.5.0 software was used for phylogenomic analysis of whole-genome sequence data [55]. MLST and cgMLST raw reads with k-mer alignment were mapped against the MLST scheme based on seven housekeeping genes, 637 locus cgMLST, and 958 locus accessory schemes (wgMLST = 1595 loci) sequences to identify the best-matched allele [56]. Phylogenetic comparison of the 53 *C. jejuni* genome sequences was performed using a neighbor-joining (NJ) tree based on a distance matrix of the core genomes of all isolates. The neighbor-joining tree was exported from SeqSphere+ in Newick-tree format and visualized on iTol v. 6.7.3 [57]. For comparison of phylogenetic differences, the reference genomes of *C. consicus* (ASM129846v.1), *C. hepaticus* (ASM168747v.2), *C. coli* (NCTC11366), *C. curvus* (ASM1337212v.1), *C. fetus* (CP000487.1), *C. lari* (RM2100), *C. showae* (ASM48038v.1), *C. upsaliensis* (RM3940), and *C. ureolyticus* (DSM20703) were used. The BLASTn [58] and ABRicate v. 1.0.1 [59] tools were used to screen the *C. jejuni* for potential genes encoding antibiotic resistance/virulence in the ResFinder [60], NCBI [61], ARG-ANNOT [62], CARD [63], MEGARes [64], and VFDB [65] databases. The PointFinder tool [60] was used to detect chromosomal point mutations leading to antibiotic resistance. Geneious Prime v.1.1 [66] and MEGA v.11.0.13 software [67] was used to align and compare detected point mutations.

## 4. Conclusions

Combining WGS and cgMLST techniques provided valuable insights into the mechanisms of antimicrobial resistance and the genetic relatedness of *C. jejuni* isolates originating from different sources. The cgMLST clustering of isolates into distinct groups or clusters indicated the spread of certain clones of antimicrobial-resistant *C. jejuni* within and between populations. The cgMLST clustering analysis revealed that *C. jejuni* isolates from human feces and broiler products were assigned to one cluster in our study, indicating a potential transmission route of *C. jejuni* between humans and animals, specifically broiler products. WGS data of 53 *C. jejuni* isolates revealed that some *C. jejuni* isolates, despite their origin, accumulated gene mutations in their genome, which may lead to changes in the bacterial phenotype, including resistance to antimicrobial agents. The detection of multiple gene mutations responsible for antimicrobial resistance in *C. jejuni* genomes suggests that *Campylobacters* in Lithuania have been exposed to selective pressure, such as antimicrobial use, over an extended period and may be developing new resistance mechanisms to antimicrobial agents.

## Figures and Tables

**Figure 1 ijms-24-16017-f001:**
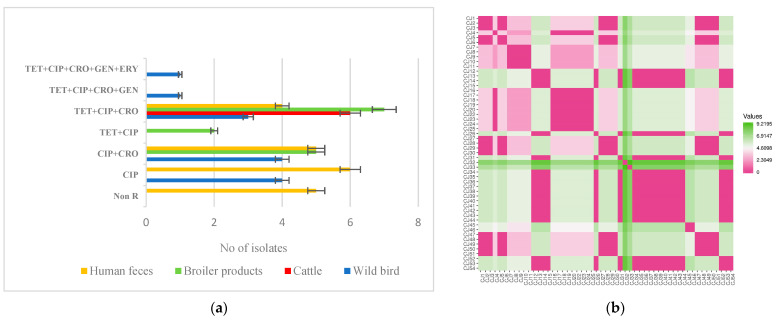
Phenotypic antimicrobial resistance patterns among different *C. jejuni* isolates (**a**). The antimicrobial resistance patterns similarity matrix heatmap of *C. jejuni* isolates (**b**).

**Figure 2 ijms-24-16017-f002:**
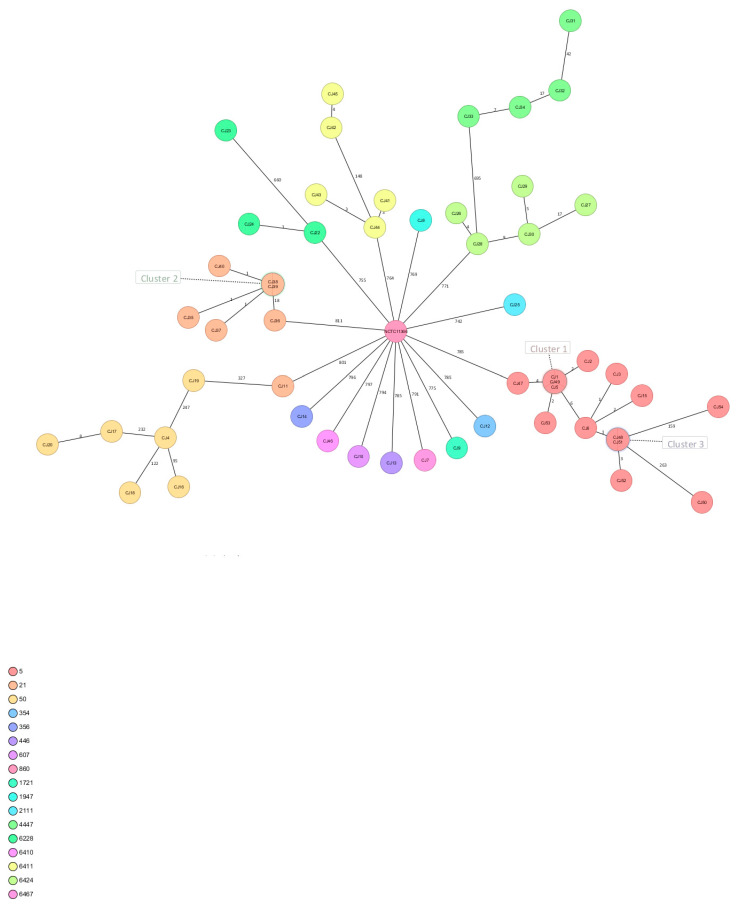
Minimum spanning tree of *C. jejuni* isolates generated using Ridom™ SeqSphere+. cgMLST analysis with numbers of allelic differences shown on connecting lines, ignoring missing values in the logarithmic pairwise scale. Each circle represents a unique cgMLST profile and is grouped by color according to the sequence type and is labeled according to the isolate name.

**Figure 3 ijms-24-16017-f003:**
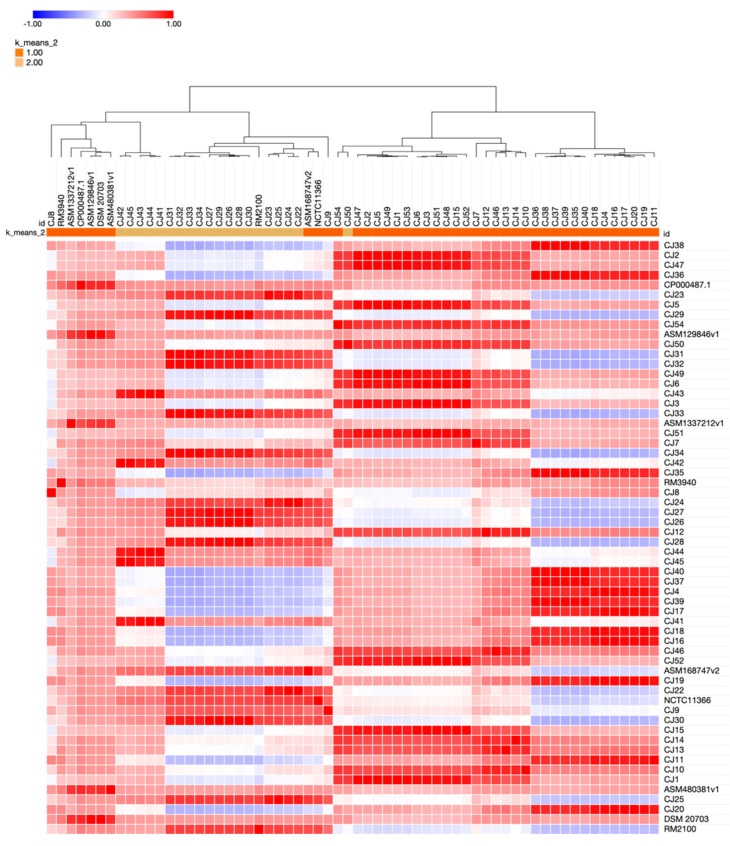
Hierarchically clustered heatmap of Jaccard’s coefficient distance matrix for *C. jejuni* isolates, with Kendall’s correlation.

**Figure 4 ijms-24-16017-f004:**
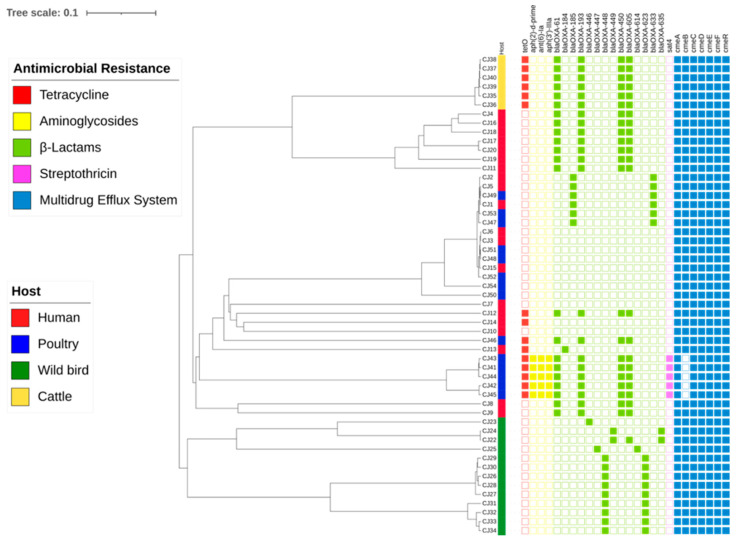
Dendrogram of hierarchical clustering and heatmap of the presence of AMR genes in 53 *C. jejuni* isolates based on WGS analysis.

**Figure 5 ijms-24-16017-f005:**
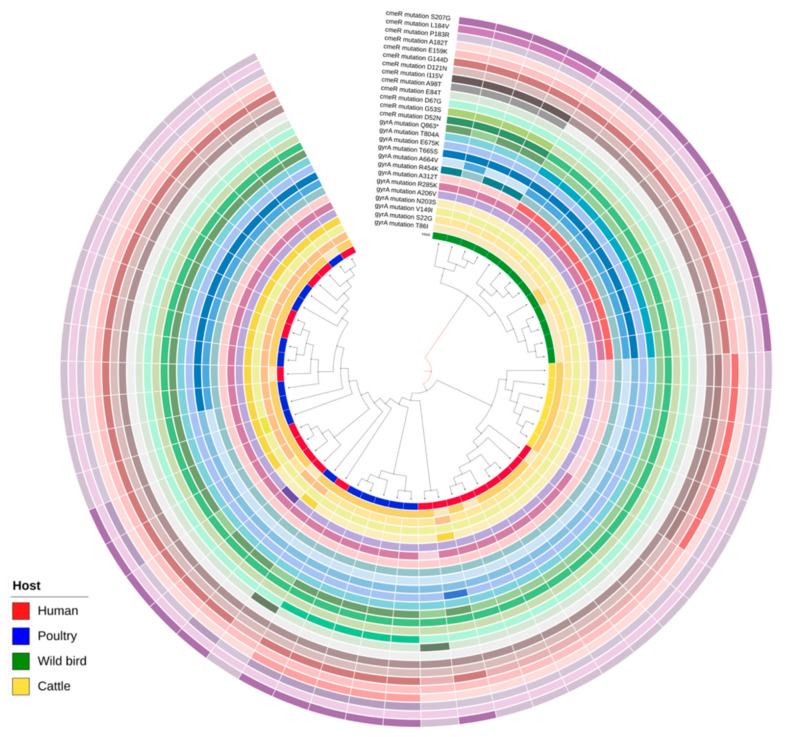
Interactive tree of life (iTOL) of 53 *C. jejuni* isolates based on the presence and absence of gene point mutations and sequence cluster classification. The darker color of the ring denotes that the mutation is present; the lighter color of the ring denotes that the mutation is absent.

**Figure 6 ijms-24-16017-f006:**
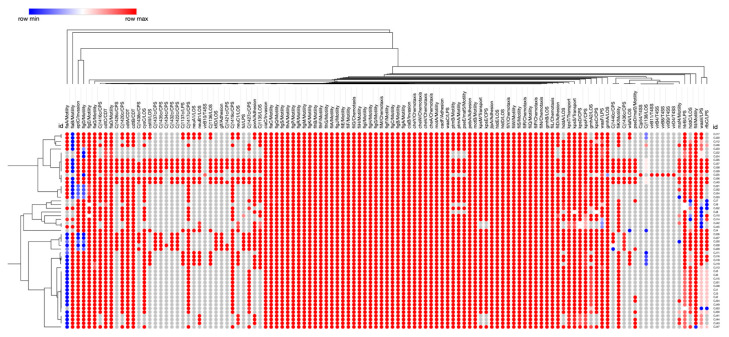
Euclidean distance clustering heatmap of virulence factors present and absent in *C. jejuni*. The red (presence = 96–100%) and blue (presence = 86–96%) circles indicate the presence of a virulence gene, whereas the gray circles (absence = 100%) indicate the absence of the virulence gene.

## Data Availability

The whole-genome sequences of the *C. jejuni* were deposited in GenBank under BioProject accession number PRJNA976194 (BioSamples SAMN35360410 to SAMN35360455).

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
