# Peer review of "Whole-Genome Sequencing-Based Profiling of Antimicrobial Resistance Genes and Core-Genome Multilocus Sequence Typing of Campylobacter jejuni from Different Sources in Lithuania"

_ijms, 2023, doi:10.3390/ijms242116017_

Round 1

Reviewer 1 Report

Comments and Suggestions for Authors

Dear Editor, Dear Authors,

the manuscript describes an interesting study on application of whole genome-based profiling on AMR genes and cgMLST of Campylobacter jejuni from different sources in Lithuania. The study showed and confirmed that combining WGS and cgMLST is an efficient approach to describe AMR mechanisms and genetic relatedness of C.jejuni isolates from different sources. Overall, the manuscript is well-written, however some minor clarifications are needed, mainly for the Result and Discussion part:

1)  based on the results, showing that >40% of C.jejuni tested were confirmed as multi-drug resistant, including dairy cattle (100%) and and broiler products (50%), could You add or expand information in the Discussion on the resistance trends in C.jejuni in the above mentioned animal populations in the recent years in the context of this study?

2) Please, add to the discussion : What is or would be the source/ reasons being behind development of extensively drug-resistant isolates found in wild birds? (Page 3, lines 109-111)?

3) Figure 2, page 4 minimum spanning tree of C.jejuni isolates and relevant reference numbers are not visible at all due to extremely small size of number. Can it be revised?

4) What was the criteria to inculde/ exclude certain C.jejuni isolates, incl from humans, cattle and wild birds in this study? How the selection of 53 isolates carried out? What wild bird species werer included, were obtained and in what conditions? Add this info in M&M!

5) Conclusion part, page 10, lines 330- please check the spelling if "campylobacters" is acceptable!

Overall, this manuscript is of good quality presenting an interesting findings on the emergence of AMR, including efficient use of both whole genome-based profiling of AMR genes as well as cgMLST  to describe AMR patterns in C.jejuni of various sources.

With this, I would suggest the manuscript for publishing after minor revision.

Reviewer #

Comments on the Quality of English Language

Some minor English review overall on manuscript is suggested.

Author Response

Dear Reviewer,

Thank you for your valuable view, and comments on our work. Your feedback has been incredibly helpful in enhancing the quality of our research. We have carefully reviewed your suggestions and revised the text according to your comments and requirements. The updated and revised version, aligned with your recommendations, can be accessed in the system. Your input has been immensely beneficial, and we greatly appreciate the time and effort you dedicated to the review process.

1)  Thank you for your suggestion. We have expanded the Discussion section to provide additional context and insights into the resistance of C. jejuni in our study, now line 106-109: In total 41.5% of the C. jejuni tested were confirmed as multidrug-resistant (MDR), especially isolated from dairy cattle (100%) and broiler products (50%). The MDR rate reported in this study showed similar results to those obtained in studies conducted in Spain [39]. However, significantly higher MDR resistance was found in studies conducted in Kenya, where MDR for C. jejuni was reported at 94.6% [40].

2)Thank you for yor valuable suggestion. In the Discussion section, we have included the following information to address this question (now line 111-118): The TET+CIP+CRO antimicrobial resistance profile was predominant and confirmed in 37.7% (Figure 1A Important, two (3.8%) C. jejuni isolates originated from wild birds were confirmed as extensively drug-resistant (XDR). Obtained data indicate that XDR C. jejuni strains circulate in wildlife and can pose a significant threat to public health, as it can be challenging and sometimes impossible to treat campylobacteriosis infections with antibiotics [41]. The possible sources could be the environmental transmission of antibiotic-resistant bacteria and resistance genes as wild birds frequently come into contact with various environmental reservoirs, including water bodies, soil, and other wildlife.[42–44].

3) Thank you for your commen tregarding Figure 2 in our manuscript, which depicts the minimum spanning tree of C. jejuni isolates. We appreciate your attention to detail and your interest in improving the visibility of reference numbers. We understand the importance of clear and accessible figures in scientific communication. However, we would like to explain the limitations we face in revising Figure 2. The original reconstruction of this figure was performed using software with limited access, which restricts our ability to make direct modifications to the figure. Given these limitations, revising the figure to enhance the visibility of reference numbers would be a time-consuming and complex process. It would require recreating the entire figure from scratch using software that allows for more precise control over the size and positioning of reference numbers. This process may lead to delays in the manuscript revision timeline and could potentially impact our ability to meet publication deadlines. To address this issue and improve the clarity of the figure, we are considering alternative solutions, such as providing a larger version of the figure. 

4) Thank you for your comment. We have clarified the information about isolates and included it in the Materials and Methods section (now line 268-270): Isolates were collected over a one-year period from humans with clinical signs consistent with campylobacteriosis and confirmed C. jejuni infections, as well as from dairy cattle, poultry products (drumsticks and wings), and wild birds, including crows and pigeons.

5) Thank you for the comment. We implementing the changes and revisions you've recommended, including the correction of 'campylobacter' to 'Campylobacter' to correctly reflect it as a genus.

Reviewer 2 Report

Comments and Suggestions for Authors

 This is a very interesting study and authors used WGS to determine ARGs/mutations in Campylobacter jejuni isolates from different sources and MLST/cgMLST to investigate the genetic relatdness among the isolates. However, authors need to make the sequencing data available - I am afraid sequencing reads are not available; https://www.ebi.ac.uk/ena/browser/view/PRJNA976194

can authors provide more details on how MLST/cgMLST was determined (lines 298-300)- which database was used?- reference or link needed.

authors need to ensure writing bacterial genus and species in italics

Author Response

Dear Reviewer,

Thank you for your valuable view, and comments on our work. Your feedback has been incredibly helpful in enhancing the quality of our research. We have carefully reviewed your suggestions and revised the text according to your comments and requirements. The updated and revised version, aligned with your recommendations, can be accessed in the system. Your input has been immensely beneficial, and we greatly appreciate the time and effort you dedicated to the review process.

1. Thank you for your comment. We understand your request for the availability of sequencing data. The C. jejuni sequences have indeed been registered and uploaded to the NCBI database with the accession number BioProject accession number PRJNA976194. However, we would like to clarify that the sequences cannot be made publicly available until the publication is officially released. This is in accordance with our publication protocol to ensure the data's integrity and proper citation. We appreciate your understanding in this matter, and once the publication is published, the sequencing data will be available: https://www.ebi.ac.uk/ena/browser/view/PRJNA976194

2. Thank you for your comment. You can find detailed information about MLSt and cgMLST (now line 307-310): Ridom™ SeqSphere+ v.8.5.0 software was used for phylogenomic analysis of whole genome sequence data [55]. MLST and cgMLST raw reads with k-mer alignment were mapped against MLST scheme based on seven housekeeping genes, 637-locus cgMLST and 958-locus accessory schemes (wgMLST=1,595 loci) sequences to identify the best-matched allele.

3. We appreciate your attention to detail. In the revised manuscript, we will ensure that bacterial genus and species names are correctly formatted in italics to adhere to the appropriate scientific nomenclature.